# Refactoring Policy for Compositional Generalizability using Self-Supervised Object Proposals

**Tongzhou Mu**[1]*    **Jiayuan Gu**[1]*    **Zhiwei Jia**[1]    **Hao Tang**[2]    **Hao Su**[1]

[1]University of California, San Diego    [2]Shanghai Jiao Tong University

{t3mu,jigu,zjia,haosu}@eng.ucsd.edu   tanghaosjtu@gmail.com

## Abstract

We study how to learn a policy with compositional generalizability. We propose a two-stage framework, which *refactorizes* a high-reward teacher policy into a generalizable student policy with strong inductive bias. Particularly, we implement an object-centric GNN-based student policy, whose input objects are learned from images through self-supervised learning. Empirically, we evaluate our approach on four difficult tasks that require compositional generalizability, and achieve superior performance compared to baselines.

## 1 Introduction

Obtaining policies that would work in different environments is a fundamental challenge of artificial intelligence. While reinforcement learning and imitation learning have made remarkable progress to solve challenging tasks (e.g., Go [31], Montezuma's Revenge [10]), their cross-environment generalizability is still limited, affecting applications in complicated and changing scenarios. In fact, to enable such applications, the policy function has to be adaptive to many factors, such as nuisances in visual appearance and physical attributes, change of rewards, and variation in the quantity and arrangement of objects.

In this paper, we aim to learn a generalizable policy network across environments composed by a flexible number of objects. To make the problem tractable, we further assume that policies of **consistency** across environments can be discovered, in the sense that a policy with consistent behavior achieves high cumulative reward following the same reasoning rationale in different environments. While this statement lacks rigor mathematically, it describes a common use case in our daily experiences: Take the classical Pacman game for example – it is not hard to write down a greedy policy that achieves high rewards, no matter how many dots and ghosts there are and where they are located.

We notice that, when talking about policy learning, we often aim to achieve two goals that are both quire challenging but not exactly aligned: 1) to address the **optimization challenge** for maximizing the reward in the training environments, which often involves high variance due to the randomness of environment dynamics and exploration strategy; and 2) to address the **generalization challenge**, which aims at achieving desired cross-environment generalizability. We conjecture that solving the two misaligned goals independently may result in two easier problems than solving the coupled problem, and there are richer options to solve each problem independently.

For this reason, we explore a strategy to break the policy learning into two stages in the compositional generalizability setup: The first stage focuses on overfitting the training environments with no generalizability concern, while the second stage focuses on designing the policy network architecture

---

Project website: https://jiayuan-gu.github.io/policy-refactorization.

to compositionally generalize the solution from the first stage. In practice, we can use the teacher-student network distillation idea to train a student network that imitates the teacher policy but has stronger generalizability. To emphasize the goal of enhancing generalizability instead of reducing network size, we call this two-stage learning strategy as **policy refactorization**, in analogy to code refactorization that aims to boost the reusability of codes. This design relieves the requirement of the first stage. We can choose networks with strong overfitting ability and fast running speed for policy optimization. For the second stage, we design or search network architectures with strong inductive-bias for generalizability.

Since the generalizability is only expected for the student network, we put more effort in investigating the performance of different student network choices (e.g., CNN, RelationNet [39], GNN). Particularly, we study GNNs with object-centric scene graphs as inputs, driven by the intuition that policy acting upon our physical world should be based on objects, attributes, and the relationship between objects, and that GNNs have strong algorithmic alignment with dynamic programming. However, unlike CNNs and RelationNet, GNNs rely on object bounding boxes to build the underlying scene graph. To make the comparison more fair and better connected with the state-of-the-art in object bounding box discovery, we evaluate GNN student policies using self-supervised object detectors learned by our improved SPACE model [21].

We empirically show that overfitting RL polices or heuristic polices can be refactorized into polices with strong compositional generalizability, in a few environments that include flexible number of objects, random object arrangement, and composition of foreground/background. Particularly, in difficult environments with sophisticated reasoning, long-range interaction, or unfamiliar background, the GNN student policy using self-supervised object detector shows the most promising results.

## 2 Approach

### 2.1 Overview

In this section, we describe our two-stage framework to learn a generalizable policy. In the first stage (Sec 2.2), our focus is to address the optimization challenge, that is, we strive to acquire a consistent teacher policy that only needs to perform well in the training environments. And this teacher policy is used to generate demonstration dataset. In the second stage (Sec 2.3), we focus on addressing the generalization challenge. We describe how we refactor the teacher policy into a student policy with strong inductive bias for generalization. Particularly, we study how to implement a GNN-based student policy that is based on a self-supervised high-recall object detector. As a by-product, attributes of objects naturally emerge in the object feature space (Sec 2.4).

### 2.2 Stage I: Demonstration Acquisition without Generalizability Concerns

In this stage, we have access to the training environment that is interactive and has image-stream based states. The outcome is a demonstration dataset, which contains state-action pairs from a policy achieving high reward in the training environment. To be more specific, we aim to generate a demonstration dataset $\mathcal{D} = \{(I_i, \pi_i)\}_{i=1}^N$, where $I_i$ is the input image (or a stack of several images) from the training environments, and $\pi_i$ is the output of demonstration policy on the input $I_i$. The representation of $\pi_i$ is flexible: it can be an action, the logits, or any latent representation, which indicates the demonstration action distribution. This dataset will be used for training both object detector and GNN-based policy.

We call the policy used to generate demonstration as *teacher policy*. It can be obtained in any way, as long as it provides reasonable and good supervision on how to solve the task, and it is not necessary to have compositional generaliziabilty. For example, one can use reinforcement learning to learn a policy, or can also use a heuristic algorithm to search for a policy. There are three caveats here. First, in our compositional generalizability setup, we usually assume that the training environments have a small number of objects, but the test environments may have many objects. For expensive optimization techniques whose complexity grow fast w.r.t. the number of objects, the setup still allows us to use them in training environments. We expect that the learned policy can scale up to the test environments. Second, besides we expect the policy to have high reward, we are also concerned about the consistency of the policy behavior across environments, and the learnability of the policy function. We do not have a deep understanding of this consistency yet; however, some examples

may be helpful to see the point. For example, in the Pacman game with no ghosts, a greedy policy that always approaches the closest dot has consistent behavior across environments, and this greedy policy is learnable by a GNN with edge convolution. More examples can be found in the experiment section. Finally, since we focus on maximizing reward rather than generalizability in the training environment, networks of high-capacity and fast running speed are favorable.

In practice, if we learn the teacher policy by reinforcement learning, we can try (or search) different network architectures (*e.g.*, CNN, GNN) and RL algorithms (*e.g.*, REINFORCE [32], DQN [24], PPO [30]), until we obtain a model solving the training environment best. The best teacher policy is used to generate the demonstration dataset by interacting with the environment.

## 2.3 Stage II: Generalizable Policy Refactorization

This stage focuses on finding or designing the student network that has good generalizability. In this work, we consider three alternatives: plain CNN, RelationNet, and GNN. Particularly, we conjecture that GNNs with object-centric scene graphs provide the strongest inductive bias for compositional generalization. To this end, we implement an effective approach to learn an object detector in a self-supervised manner (Sec 2.3.1) and use GNNs to learn policy based on imperfect proposals generated by the object detector (Sec 2.3.2).

### 2.3.1 Obtaining High-Recall Object Proposals

In this paper, we build a self-supervised object detector based upon SPACE [21]. Different from other self-supervised object detection methods based on reconstruction [11, 6] or motion [18, 15], SPACE is able to detect salient objects from relative complex backgrounds. It is self-supervised by reconstructing a single image with foreground objects and background stuff. However, it is sometimes unstable to train and sensitive to several hyperparameters, *e.g.*, prior object sizes [22]. Therefore, we improve SPACE by introducing a better parameterization for bounding box regression, which is widely used in supervised object detection [14]. More details can be found in the supplementary.

The reason to use an unsupervised object detector is that it requires no extra labeling, and can be easily applied to widely used environments, *e.g.*, Atari [2] and ProcGen [5]. However, note that our framework is not designed for any specific object detector. The only expectation is that the recall should be relatively high with a reasonable precision, which is a practical assumption.

### 2.3.2 Learning GNN-based Policy with Objects from Demonstration

The desired compositional generalizability of policy in this work is indeed the consequence of having an underlying reasoning mechanism (i.e., the program to generate a feasible policy) that is invariant to the composition of objects in environments. For example, in a Pacman game, policies with reasonable performance may be derived based on the reasoning of approaching a close dot (food) and escaping nearby ghosts, applicable to different amount of food and ghosts. To achieve such compositional generalizability, we need a tool that has the representation power of capturing such underlying reasoning mechanism (*e.g.*, a greedy algorithm). Under the assumption that the task is relevant to objects and the relationship between them, we employ graph neural networks to represent the policy based on the detected objects, due to the impressive power of GNN in reasoning, including approximating dynamic programs in certain scenarios [38].

**Behavior Cloning by GNN**  Given the demonstration dataset $\mathcal{D}$, we want to learn a GNN-based policy via behaviour cloning [27]. Concretely, we minimize the following loss: $L = \sum_{(I_i, \pi_i) \in \mathcal{D}} \|f_{\text{GNN}}(\mathcal{O}_i, I_i) - \pi_i\|_2^2$, where for each image $I_i$ in the training set, $\mathcal{O}_i$ corresponds to the bounding boxes generated by the object detector, and $f_{\text{GNN}}$ denotes the policy GNN to be trained. The input to our GNN-based policy is an object-centric graph. The nodes are detected objects and the node features are the image patches cropped from the original image based on the object bounding boxes provided by the object detector. And we further encode them into latent features by a CNN. As for the edges between the nodes, we can adopt different types of graph structures depending on the type of the task, e.g., complete graph or empty graph (no edges). The design of our policy GNN architecture will be elaborated in Sec 3 in a task-specific manner.

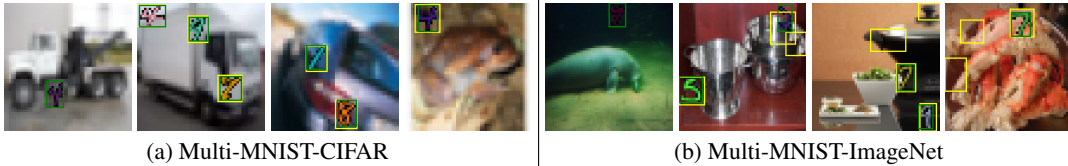

|(a) Multi-MNIST-CIFAR|(b) Multi-MNIST-ImageNet|

Figure 1: Examples of detection results in *ascending order w.r.t their data parameters* $\sigma_i$. GT bounding boxes are annotated in green and proposed boxes are in yellow. The leftmost image has the smallest data parameter and is down-weighted for learning, since the only digit is missing. The rightmost image has the largest data parameter and is easiest to learn, since it contains only one digit.

**Down-weighting Demonstrations with Incomplete Task-Relevant Object Detection**   Due to the imperfectness of object detector, chances are that some task-relevant objects are missing and task-irrelevant objects are included. We observe that GNN is robust to irrelevant objects, but is sensitive to incomplete proposals. It is sometimes impossible to solve certain tasks when any object is missing. Those bad data samples with incomplete proposals may prevent the policy GNN to learn a good policy as they force the network to sacrifice the generalizability to fit out-of-distribution data.

To this end, we introduce data parameters [29] to downweight data samples with incomplete object proposals. Different from [29], we apply data parameters to a general loss rather than cross-entropy loss. Concretely, we associate a data parameter $\sigma_i$ with each data sample $x_i$. Within a batch, we reweight the total loss according to data parameters: $L = \sum_{i=1}^{b} \frac{e^{\sigma_i}}{\sum_{j=1}^{b} e^{\sigma_j}} l(x_i)$ , where $l(x_i)$ is the loss calculated on the data sample $x_i$. It can be also considered as a deep version of RANSAC [12].

## 2.4   Task-Relevant Knowledge Discovery of Objects and Attributes

One advantage of our framework is that it supports interpretable model diagnosis, thanks to the modular design and explicit object modeling. The task-relevant feature is learned for each object by our policy GNN. We can apply K-means clustering or other data mining techniques to learned object features to discover task-relevant knowledge. Object attributes emerge through the clustering process, and other post-processing can be applied to remove task-irrelevant objects or label data by clustering results. We show some qualitative results in Sec 3.

## 3   Experiments

We experiment on four different environments to evaluate our proposed methods. We start from evaluating the basic units, the SPACE object detector and the object-centric GNN on Multi-MNIST. After the units are verified, then, we evaluate the effectiveness of our framework for two types of compositional generalizability: w.r.t. the change of object quantity (FallingDigit), and w.r.t. the change of background (BigFish). Finally, we show that there exist environments, *e.g.*, Pacman, in which a generalizable student policy does not have to be object-centric GNNs.

### 3.1   Multi-MNIST

**Task Description**   We first introduce a single-step task called Multi-MNIST. Given a 54x54 image on which some MNIST digtis are scattered randomly, the task is to calculate the sum of the digits. The training set consists of 60000 images and each image has 1 to 3 MNIST digits, while the the test set consists of 10000 images with 4 MNIST digits. The task is inspired by AIR [11], but we render digits with random colors on complex backgrounds from two different sources: CIFAR-10 [19] and ImageNet [8]. The datasets generated are denoted by Multi-MNIST-CIFAR and Multi-MNIST-ImageNet, respectively. Examples images are shown in Fig 1.

**Method Details**   In this task, we train all the baselines in a supervised learning manner, but it is equivalent to train a policy by REINFORCE [32] with appropriate rewards. Since this task is irrelevant to the relationship between objects, we use an empty graph (without edges) as the object-centric graph. The node input is a patch cropped from the image according to the bounding box of the

| Method | Train Acc | Test Acc |
|---|---|---|
| CNN | 92.0(1.7) | 30.7(4.9) |
| Relation Net | 97.4(0.5) | 18.2(13.3) |
| GNN+SPACE (ours) | 83.1(0.1) | **51.2(3.8)** |
| GNN+GT boxes | 99.5(0.1) | 81.4(2.0) |
| GNN+SPACE (w/o DP) | 86.1(0.6) | 29.1(3.6) |
| CNN (with DP) | 86.9(1.2) | 49.8(2.7) |

(a) Multi-MNIST-CIFAR

| Method | Train Acc | Test Acc |
|---|---|---|
| CNN | 90.5(2.9) | 12.0(2.1) |
| Relation Net | 96.4(0.8) | 8.4(4.7) |
| GNN+SPACE (ours) | 80.2(0.2) | **51.2(1.2)** |
| GNN+GT boxes | 99.1(0.1) | 69.8(10.5) |
| GNN+SPACE (w/o DP) | 80.4(0.6) | 27.7(1.3) |
| CNN (with DP) | 83.7(2.7) | 14.2(2.1) |

(b) Multi-MNIST-ImageNet

Table 1: Quantitative results on Multi-MNIST. The average with the standard deviation (in the parentheses) over 5 trials is reported. *DP* stands for data parameters introduced in Sec 2.3.2.

corresponding object and resized to 16x16. Then we use a CNN to encode node features, and apply a global-add-pooling to readout a global feature over all the nodes, followed by an MLP to predict the summation. And the policy GNN is implemented as PointNet [26].

We compare our method with two baselines: plain CNN and Relation Net [39]. For the two baselines, we flatten the feature map generated by the convolution layers and apply an MLP to predict the summation. It seems to be the best design choice according to our preliminary attempts. Other design choices are discussed in the supplementary. Besides, we apply the *RandomResizeCrop* to images as the data augmentation for all baselines and our model.

**Results** Table 1 shows the quantitative comparison between our GNN-based policy and two baselines, as well as ablation studies. To measure the performance of the task, we report the accuracy, where the prediction is correct if its absolute difference with the ground truth is less than 0.5. The recall/AP@0.25 of the object detectors on Multi-MNIST-CIFAR and Multi-MNIST-ImageNet are 93.7/92.4 and 93.8/31.5 respectively. Our GNN-based policy outperforms two baselines by a large margin, which shows the advantage of our method over CNN-based method w.r.t compositional generalization. Our policy GNN trained with ground truth boxes achieves high accuracy (81.4 and 69.8), which verifies our implementation. However, it is still imperfect, which implies the inherent difficulty of this task. Besides, our policy GNN fails to generalize well without data parameters. The effect of data parameters is analyzed in Fig 1.

**Analysis** One advantage of our framework is that it supports interpretable model diagnosis, thanks to the modular design and explicit object modeling. First, object attributes emerge when the learned object features are clustered. Fig 2 visualizes the learned object features by t-SNE [23]. For each detected object, we label it with the class of its closet ground truth object if the overlap is larger than 0.25, otherwise it is labeled as background. It is observed that task-driven object features are more distinguishable compared to reconstruction-driven ones.

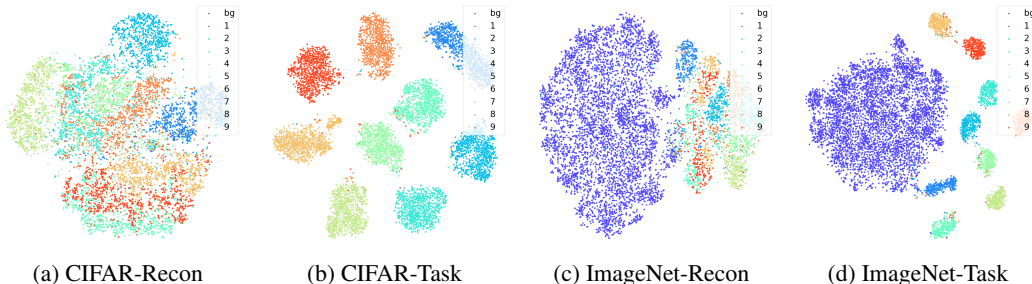

(a) CIFAR-Recon     (b) CIFAR-Task     (c) ImageNet-Recon     (d) ImageNet-Task

Figure 2: t-SNE [23] visualization of the learned object features by SPACE [21] and our policy GNN. For CIFAR-Recon, the object features are the latent features used in SPACE [21] to reconstruct foreground objects on Multi-MNIST-CIFAR. For CIFAR-Task, the object features are node features encoded by CNN used in our policy GNN to solve the task on Multi-MNIST-CIFAR.

### 3.2 FallingDigit

**Task Description**  Inspired by the classical video game "Tetris", we design this FallingDigit game. In this game, there is a digit falling from the top and some target digits lying on the bottom. The player needs to control the falling digit to hit the closest target digit (in terms of digit value). At each step, the falling digit can be moved to one of the three direction: down-left, down, down-right. The player receives +1 reward when the falling digit hits the correct target, and the target digit will be cleared. The player receives -1 reward when it hits the bottom or wrong targets, and the wrong targets will not be cleared. The falling digit will disappear when it hit bottom or any target digits, and a new falling digit will be created at the next timestep. The positions of all digits are random. An episode will terminate when all the target digits are cleared or the player has taken 100 actions. The training environment has 3 target digits, while the agent is tested in the environments with more target digits. To generalize, the agent has to localize and identify numbers from pictures, learn to compare the difference between digits, and handle interactions between distant pixels. We create two variants of this game: FallingDigit-Black (background is black) and FallingDigit-CIFAR (background is selected from a subset of CIFAR). Digits are from MNIST. Examples are shown in Fig 3.

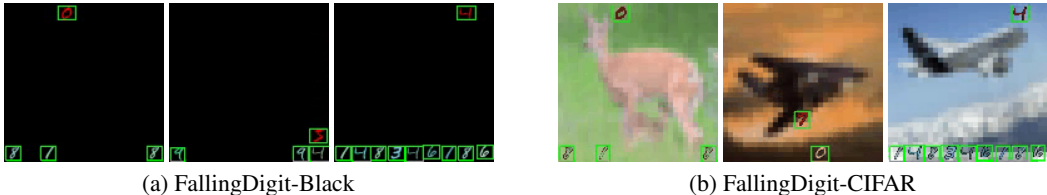

(a) FallingDigit-Black                    (b) FallingDigit-CIFAR

Figure 3: Examples of FallingDigit games with different backgrounds. Object proposal generated by our improved SPACE are annotated in green bounding boxes.

**Method Details**  For our framework, we first train a teacher policy by DQN [24] in the training environment, which can converge to a near-optimal solution. The architecture of teacher policy is Relation Net [39]. The teacher policy is used to collect 60000 images through the interaction with the environment, and label them with the taken actions, as illustrated in Sec 2.2. When building the demonstration dataset, we filter out a few episodes with low rewards so as to avoid providing incorrect demonstrations. Then, we train the self-supervised object detector on the collected images, and train the policy GNN based on the generated object proposals. Since it is critical to reason about the relationship between the falling digit and other target digits in this game, we use a complete graph as the object-centric graph. The node input includes the bounding box position and a patch cropped from the image according to the bounding box, which is resized to $16 \times 16$. The policy GNN is implemented as EdgeConv [36]. We compare our method with two baselines: plain CNN and Relation Net [39], both of them are trained by DQN algorithm. To make a fair comparison, the RL agents are trained on a fixed set of episodes, which are also the source of the demostration dataset.

**Results**  Fig 4 shows the mean episode rewards of different methods in the environment with different number of target digits. In all the environments, our agent outperforms the Relation Net (which is our RL teacher) and CNN baselines. Our agent can generalize well to the environments with 9 target digits, while the performance of Relation Net and CNN gradually decrease when the number of objects is increasing. We also generate the ground truth action labels by a heuristic program, which can serve as the teacher policy. It results in even better generalizability than the RL teacher, since it provides perfect and more consistent demonstration.

### 3.3 BigFish

**Task Description**  BigFish is a game from ProcGen Benchmark [5], which provides a challenging generalization scenario to RL agents. Fig 5 shows some examples of the game. In this game, the player needs to eat fish smaller than itself to gain rewards and make itself larger. Contacting with a larger fish will terminate the episode. To evaluate the compositional generalizability w.r.t different backgrounds, we modify the BigFish game to create new test environments with unseen complicated

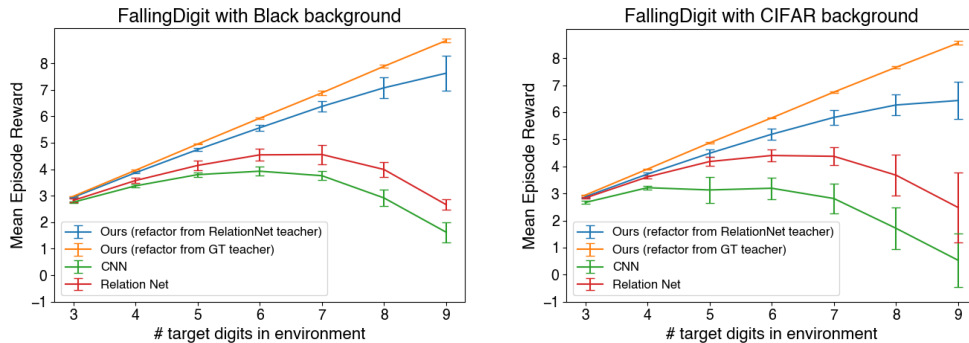

Figure 4: Quantitative results of generalizing to the FallingDigit environments with different number of target digits. Ours uses GNN+SPACE architecture. CNN and Relation Net are trained by DQN. For ours (refactor from Relation Net teacher), we report the mean episode reward over 9 different runs (3 different RL teacher runs and 3 refactorization runs per teacher). For the other baselines, the result is averaged over 3 runs. The error bar shows standard deviation.

backgrounds. In our experiment, we train agents on level 0-199 and zero-shot test on level 500-599 with unseen backgrounds. The difficulty mode of the game is set to easy.

**Method Details**    Following the baseline provided by Procgen Benchmark [5], we use PPO[30] to train a CNN-based policy network. When collecting the demonstration dataset, we use $\epsilon$-greedy exploration strategy to increase the diversity of states. And we apply data augmentation by adding some object proposals with low confidence scores. The details can be found in the supplementary. Each image in the demonstration dataset is labeled with the softmax logits of action by the teacher policy. Since the task requires reasoning about the relation between the player and other fishes, we also use a complete graph. The architecture details of our policy GNN are in supplementary. We compare our method against CNN and Relation Net, both are trained by PPO for 200M frames. And the CNN serves as the teacher in our framework.

**Results**    Table 2 shows that our refactorized GNN-based policy outperforms the two RL baselines in the test environments with unseen complicated backgrounds, which shows the object-centric graph as inductive bias makes the policy more robust to background changes. In the training environment, the CNN gets 29.65 and the Relation Net gets 28.22. Since our GNN-based policy is trained with augmented low-confidence object proposals, it performs slightly worse than the CNN teacher in the training environment (gets 24.86).

| Method | Test on unseen backgrounds |
|---|---|
| CNN | 4.40(1.90) |
| Relation Net | 4.54(1.22) |
| Ours (refactor from CNN) | **6.05(2.44)** |

Table 2: Quantitative results on BigFish. Ours uses GNN+SPACE architecture. CNN and Relation Net are trained by PPO. For ours, we report the mean episode reward over 24 different runs (4 different RL teacher runs, 3 different demonstration datasets per teacher and 2 refactorization runs per dataset). For the other baselines, the result is averaged over 4 runs. The standard deviation is in the parentheses.

## 3.4   Pacman

**Task Description**    We build a customized Pacman game based on [3]. Fig 6 shows an example. The objective of this game is to control the Pacman to eat all the dots. The initial positions of all objects are random at each episode. At each time step, the agent can take one of the four actions, to move one step towards left, right, up or down in a 14x14 grid. The agent receives +1 point as the

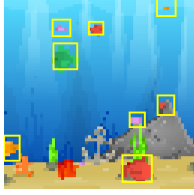 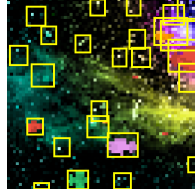 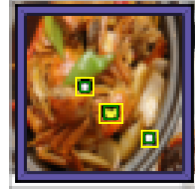

Figure 5: Examples of BigFish games and object proposals generated by SPACE (annotated in yellow boxes.) The left one is sampled from training environments and the right one is from test environments.

Figure 6: Examples of our customized Pacman game. GT bounding boxes are in green, object proposals are in yellow.

reward for eating a dot, and -0.01 point for each move. An episode will terminate when all the dots are eaten or the agent takes 100 actions. The training environment is initialized with 2 dots, while the agent is tested in the environments initialized with more dots. The backgrounds in this game are random selected from a subset of ImageNet.

**Method Details**  The basic experimental method is similar to FallingDigit in Sec 3.2. In this experiment, we focus on comparing the different student network architectures. We use a CNN-based policy as the teacher since it achieves almost perfect scores in the training environments. Then we refactor it into obejct-centric GNN, CNN and Relation Net. The architecture details of each student can be found in supplementary materials.

| Method | 2 | 3 | 5 | 10 |
|---|---|---|---|---|
| CNN (RL) | 1.86(0.02) | 2.63(0.14) | 1.64(0.31) | -0.41(0.09) |
| Relation Net (RL) | 1.84(0.04) | 2.76(0.08) | 4.31(0.16) | 4.40(0.32) |
| CNN (refactor) | 1.86(0.00) | 2.80(0.00) | 4.70(0.01) | **9.17(0.09)** |
| Relation Net (refactor) | 1.86(0.00) | 2.80(0.01) | 4.70(0.00) | **9.11(0.23)** |
| GNN+SPACE (refactor) | 1.86(0.02) | 2.80(0.02) | 4.63(0.08) | **8.32(0.31)** |

Table 3: Quantitative results on Pacman. The mean episode reward with the standard deviation (in the parentheses) over three different runs is reported. *RL* and *refactor* indicates the ways we train this architecture.

**Results**  Table 3 shows the mean episode rewards of different methods in the environments with different number of dots. The refactorized GNN policy generalizes well to the environments with 10 dots, and outperforms all RL baselines. However, in this Pacman environment, the refacorized CNN and refactorized Relation Net also generalizes very well to the environments with 10 dots. Note that in the three other environments mentioned above, the GNN student performs better than CNN and Relation Net students according to our experiments (see project website for more results). This shows that, for some environments like Pacman, a generalizable student policy does not have to be object-centric GNNs. We feel that the main reason for the effectiveness of CNN and Relation Net in this environment is because it could make reasonable decisions just by looking at nearby regions. Another observation is that the refactorized CNN student generalizes better than its CNN teacher, but this phenomenon is not observed in other three environments.

## 4 Related Work

**Structured RL**  Many works have investigated structured representations and structured policies in the RL literature. [17, 1, 7, 13] show that structured polices induced by structured representations(*e.g.*, symbols, objects and scene graphs) generalize better than unstructured counterparts. Other works explicitly model the structure of an agent, and use GNN to learn either a policy [35] or a forward model [28]. However, they rely on ground truth object(part) information or graph structures. Another series of works [39, 25] make use of attention mechanism to augment model-free deep RL agents, which improves performance, learning efficiency, generalization and interpretability. However, attention maps can not provide as interpretable analysis as our framework.

[34] proposes an object-centric perception approach to deep control problems, which shows better generalizability and interpretability. Our method differs from it in several aspects: 1) our demonstration dataset is generated by RL algorithms while their dataset is annotated by human; 2) they do not include GNN to reason about objects; 3) our object detector is unsupervised while their object detector is trained with supervision.

**Self-Supervised Object Discovery**   Our object detector is built upon a self-supervised object detector, SPACE [21]. It extends [11, 6] and decomposes an image into salient foreground objects and background stuff, self-supervised by reconstruction.

Several works attempt to combine self-supervised scene decomposition approaches with model-based RL. COBRA [37] employs MONet [4] to obtain object latent representations, which are used to train an exploration policy. OP3 [33] extends IODINE [16] and applies a planning module on top of learned entities. Their objects are represented by latent slots. Both work show the ability to generalize to novel tasks; however, they suffer from the drawback of scene decomposition approaches and are not able to handle multiple instances of the same category (shape and color), thus cannot be used to solve our experiment environments. Other approaches rely on motion clues. [15] learns moving object segmentation in an unsupervised fashion, which helps improve sample efficiency. [9] makes use of a video prediction model capable of capturing object dynamics to achieve faster convergence and better generalization. [20] learns unsupervised keypoint detection, and uses both the keypoint co-ordinates and corresponding image features to improve sample efficiency. Different from our method, they augment CNNs with either features or outputs from their perception modules, rather than directly learn policies from detections.

## 5   Discussion and Future Work

Our two-stage scheme allows us to decouple the efforts to address the optimization challenge and the generalization challenge. We observe that, with our SPACE empowered object-centric GNNs, we can refactor a high reward policy learned in the training environments into a student policy with better generalizability. Practically, this scheme also improves training speed, *e.g.*, training a GNN by RL with online-generated object proposals is super slow, but training a CNN teacher and refactoring it into a GNN is faster. While our experiments have been focusing on compositional generalizability, we feel that this scheme may benefit other types of generalizabilities, like variation in visual appearance and reward function, as well.

Besides the overall framework, we also improved the SPACE method, which learns a self-supervised object detector to be used by GNN. Adding the object proposal module makes the framework closer to a white-box system. Along with the learning process, we can also discover task-relevant objects and attributes. The human readable representation, including the objects and attributes, can benefit diagnosis of the algorithm, result interpretation, and knowledge increment for transfer learning.

Some of the design choices in this work are a double-edged sword: 1) We assume the accessibility of a high-recall object proposal algorithm. Compared with more end-to-end methods, such as plain CNN-based RL [24] or attention-based relational reasoning framework [39], the object proposal algorithm serves as a strong inductive bias, which may fail the policy learning when the inductive bias is inappropriate; 2) We rely on the Graph Neural Network (GNN) to achieve compositional generalizability. The limitations of the GNN toolkit would also restrict the power of our approach. Nonetheless, the research into more powerful GNNs is a hot topic and the technique is improving; 3) Our policy is object-centric but has ignored the important role stuffs play for reasoning.

Finally, this work has left many open questions, even if we have explored to the best of our endeavours. First, while we are aware that policy consistency and learnability are important, we lack rigor definition and analysis to this issue. Second, our understanding to how the policy refactorization would work is quite limited. A particularly interesting case is that, policy refactorization tends to change the generalizability of a teacher policy, even if the student policy uses exactly the same architecture (*e.g.*, CNN in Pacman could generalize much better as a student than as a teacher, but RelationNet in FallingDigit generalized slightly worse as a student than as a teacher). We conjecture it is relevant to the property of the network, the game, and the order that data are fed. We leave all these open questions to future work.

# 6 Acknowledgement

This research was supported by NSF grant IIS-1764078. We also acknowledge Sirui Xu for his help in discussions and experiments.

# 7 Broader Impact

Our work is a basic step towards building autonomous agents that can train in limited environment and perform well in more complicated environment with similar reasoning rationale, using vision as the primary information source. Particularly, we try to build a system that makes decisions in a way that human may interpret. From an ethical aspect, it is helpful to build AI that humans can better communicate with.

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
