[Supplementary Material]

# Supplementary Material of
# Refactoring Policy for Compositional Generalizability using Self-Supervised Object Proposals

**Tongzhou Mu**[1*]     **Jiayuan Gu**[1*]     **Zhiwei Jia**[1]     **Hao Tang**[2]     **Hao Su**[1]
[1]University of California, San Diego
{t3mu,jigu,zjia,haosu}@eng.ucsd.edu
[2]Shanghai Jiao Tong University
tanghaosjtu@gmail.com

## 1   Overview

This supplementary material includes implementation details relevant to network architectures and hyperparameters, as well as additional experiments to analyze the robustness of our two-stage framework. Sec 2 illustrates our improvements made to SPACE [12]. Sec 3, 4, 5, and 6 describe the network architectures and hyperparameters used in our experiments on Multi-MNIST, FallingDigit, BigFish, and Pacman, respectively. A robustness analysis is presented in Sec 7, where we also show the advantages of our two-stage framework over end-to-end training object-centric GNNs by RL.

## 2   Improvements of SPACE

In this section, we provide a recap of SPACE [12] along with the improvements. We refer readers to the original paper for more complete explanation. SPACE is a unified probabilistic generative model that combines both the spatial-attention (foreground) and scene-mixture (background) models. It assumes that an image (or scene) can be decomposed into foreground and background latents: $z^{fg}$ and $z^{bg}$. The foreground $z^{fg}$ consists of a set of independent foreground objects $z^{fg} = \{z_i^{fg}\}_{i=1}^N$. The observed image $x$ is modeled as a sample from the pixel-wise Gaussian mixture model, which is a combination of foreground and background image distributions, as illustrated in Eq 1.

$$p_\theta(x|z^{fg}, z^{bg}) = \alpha p_\theta(x|z^{fg}) + (1-\alpha)p_\theta(x|z^{bg}) \tag{1}$$

, where $\theta$ is the parameters of the generative network, $\alpha$ is the foreground mixing probability.

**Foreground**   The foreground image component is modeled as a Gaussian distribution $p(x|z^{fg}) \sim \mathcal{N}(\mu^{fg}, \sigma^{fg})$. It is represented as structured latents. Concretely, the image is divided into $H \times W$ cells, and each cell represents a potential foreground object. Each cell is associated with a set of latent variables $\{z_i^{pres}, z_i^{where}, z_i^{depth}, z_i^{what}\}$. The underlying idea is similar to RCNN [18] and YOLO [17]. $z^{pres} \in \mathcal{R}$ is a binary random variable indicating the presence of the object in the cell. $z^{what}$ models the object appearance and mask, and $z^{depth} \in \mathcal{R}$ indicates the relative depth of the object. $z^{where} \in \mathcal{R}^4$ parameterizes the object bounding box. For each cell that $z_i^{pres} = 1$, SPACE uses $z_i^{what}$ to decode the object reconstruction and its mask. The object reconstruction is positioned on a full-resolution canvas using $z_i^{where}$ via the Spatial Transformer Network [8]. The object depth $z_i^{depth}$ is then used to fuse all the object reconstructions into a single foreground image reconstruction $\mu^{fg}$. In practice, $\sigma^{fg}$ is treated as a hyperparameter.

**Improvement: Bounding Box Parameterization**   As illustrated in [13], SPACE [12] is somehow unstable to train and sensitive to some hyperparameters, *e.g.*, prior object sizes. In the original paper,

$z^{where}$ is decomposed into two latent logits $z^{shift}$ and $z^{scale}$, activated by $tanh$ and $sigmoid$, which represent the shift and scale of the bounding box respectively. Thus, to change the prior size of objects, it is required to tune those non-intuitive logits. Besides, the scale $z^{scale}$ is relative to the whole image instead of the cell, which makes the model more fragile.

In our implementation, following R-CNN [18], we reparameterize the bounding box $(x_{ctr}, y_{ctr}, w, h)$ as an offset $(dx, dy, dw, dh)$ relative to a pre-defined anchor $(x_a, y_a, w_a, h_a)$ centered at the cell. For simplification, we associate each cell with only one anchor. It can be easily extended to multiple anchors per cell [18] or multiple levels [11]. Eq 2 illustrates the formula of the 'bounding-box regression' reparameterization.

$$
\begin{aligned}
dx &= (x_{ctr} - x_a)/w_a \\
dy &= (y_{ctr} - y_a)/h_a \\
dw &= \log((w - w_a)/w_a) \\
dh &= \log((h - h_a)/h_a)
\end{aligned}
\tag{2}
$$

Compared to the original implementation, ours supports more intuitive interpretation of the hyperparameters related to the bounding box. The priors of $(dx, dy, dw, dh)$ can be simply modeled as zero-mean Gaussian distributions. And the variances of Gaussian distributions can be tuned to control how much the bounding box can be different from the anchor.

**Improvement: Background**    In the original paper [12], the background is modeled as a sequence of segments by GENESIS [4]. From our experiments, we find that the key to design a background module is its capacity. Thus, it is not necessary to use complex and expensive models, *e.g.*, MONet [2], GENESIS [4], IODINE [7], especially for the RL applications. It is even not necessary to use a VAE. By using a much simpler background module, our improved SPACE model can be easily trained with a single GPU and a higher, unified learning rate, which results in faster convergence. In contrast, the default configuration of SPACE requires 4 GPUs to train, separate learning rates for different modules, and other sophisticated tricks (fixed $\alpha$ at the beginning of training).

**Inference and Training**    All the random variables follow Gaussian distributions, except for $z^{pres}$ follows a Concrete distribution [14]. As the model is a variational autoencoder (VAE), the reparameterization trick [9] is used to optimize the ELBO.

## 3  Multi-MNIST

### 3.1  Self-supervised Object Detector

SPACE [12] consists of several modules: foreground image encoder, glimpse encoder, glimpse decoder, background image encoder, background image decoder. We refer readers to [12] for detailed explanation of each module. For the glimpse of each object, we apply the spatial transformer network [8] (STN) to crop a patch from the image according to the bounding box and resize it to 14x14. Table 1 shows the architecture of the object detector used in the experiments. Table 2 shows the hyperparameters of the object detector.

### 3.2  Policy GNN

**Network Architectures**    In the Multi-MNIST experiment, the policy GNN is implemented as PointConv [20] in Pytorch Geometric [6]. The input graph is an empty graph (without edges). Each node corresponds to a detected object and the node feature is an embedded image feature $x_{img}$. We apply the STN to crop a patch from the image according to the bounding box and resize it to 16x16, and then we encode the patch by a CNN to get $x_{img}$. The CNN encoder is denoted by *Image Patch Encoder*. Note that the bias terms are removed in the fully connected layers after the global readout function. Table 3 shows the architecture of GNN and image patch encoder used in Multi-MNIST experiments.

**Hyperparameters in Training**    When training the GNN, the batch size is 64. The initial learning rate is 0.001, and is divided by 2 every 100K gradient updates. The network is trained with the Adam optimizer for 500K gradient updates.

**Foreground Image Encoder**

| Layer | Resolution | Stride | Norm./Act. |
|---|---|---|---|
| Input | 54x54x3 | | |
| Conv 3x3 | 54x54x64 | 1 | BN/ReLU |
| Conv 3x3 | 18x18x64 | 3 | BN/ReLU |
| Conv 3x3 | 18x18x128 | 1 | BN/ReLU |
| Conv 3x3 | 6x6x128 | 3 | BN/ReLU |
| Conv 3x3 | 6x6x256 | 1 | BN/ReLU |
| Conv 1x1 | 6x6x128 | 1 | BN/ReLU |
| Conv 1x1 | 6x6x128 | 1 | BN/ReLU |
| | 6x6x1 (object presence $z^{pres}$) | 1 | Sigmoid |
| Conv 1x1 | 6x6x4 (bounding box mean $z^{where}$) | 1 | |
| | 6x6x4 (bounding box stdev $z^{where}$) | 1 | Softplus |

**Glimpse Encoder**

| Layer | Resolution | Norm./Act. |
|---|---|---|
| Input | 14x14x3 | |
| Flatten | 588 | |
| Linear | 256 | GN(16)/ReLU |
| Linear | 256 | GN(16)/ReLU |
| Linear | 50 (mean $z^{what}$) | |
| | 50 (stdev $z^{what}$) | Softplus |

**Glimpse Decoder**

| Layer | Resolution | Norm./Act. |
|---|---|---|
| Input | 50 | |
| Linear | 256 | GN(16)/ReLU |
| Linear | 256 | GN(16)/ReLU |
| Linear | 784 | Sigmoid |
| Reshape | 14x14x4 | |

**Background Image Encoder**

| Layer | Resolution | Stride | Norm./Act. |
|---|---|---|---|
| Input | 54x54x3 | | |
| Conv 3x3 | 54x54x64 | 1 | BN/ReLU |
| Conv 3x3 | 18x18x64 | 3 | BN/ReLU |
| Conv 3x3 | 18x18x128 | 1 | BN/ReLU |
| Conv 3x3 | 6x6x128 | 3 | BN/ReLU |
| Conv 3x3 | 6x6x256 | 1 | BN/ReLU |
| Maxpool | 256 | | |

**Background Image Decoder**

| Layer | Resolution | Norm./Act. |
|---|---|---|
| Linear | 256 | BN/ReLU |
| Linear | 256 | BN/ReLU |
| Linear | 8478 | Sigmoid |
| Reshape | 54x54x3 | |

Table 1: The architecture of the self-supervised object detector for all the experiments on Multi-MNIST.

## 3.3 Baselines

**Network Architectures**   Table 4 shows the architecture of the plain CNN used in the experiments on Multi-MNIST. The Relation Net shares the same CNN backbone with the plain CNN. For the Relation Net, following [21], we add a relation module after the final feature map by a residual connection. The architecture of the relation module is illustrated in Table 5. For these baselines, the last CNN feature map is flattened and a multi-layer perceptron (MLP) is applied to get the final output. Other design choices, *e.g.*, max pooling over the last feature map, are investigated and the quantitative results are shown in Table 6.

**Hyperparameters in Tranining**   All the baseline CNNs are trained with the Adam optimizer for 500K steps. The initial learning rate is also 0.001, and is divided by 2 every 100K steps.

| Name | Value | Schedule |
|---|---|---|
| max iteration | 100K | |
| optimizer | Adam | |
| batch size | 64 | |
| learning rate | 1e-3 | |
| gradient clip | 1.0 | |
| $z_{pres}$ prior | $0.1 \rightarrow 0.01$ | $10K \rightarrow 50K$ |
| $z_{pres}$ temperature | $2.0 \rightarrow 0.1$ | $10K \rightarrow 50K$ |
| $z_{where}$ prior mean | 0 | |
| $z_{where}$ prior stdev | 0.2 | |
| $z_{what}$ prior mean | 0 | |
| $z_{what}$ prior stdev | 1.0 | |
| $z_{what}$ dimension | 50 | |
| $z_{depth}$ prior mean | 0 | |
| $z_{depth}$ prior stdev | 1.0 | |
| $z_{depth}$ scale | 10.0 | |
| fg recon prior stdev | 0.15 | |
| bg recon prior stdev | 0.15 | |

Table 2: The hyperparameters of the self-supervised object detector for all the experiments on Multi-MNIST.

**Image Patch Encoder**

| Layer | Resolution | Stride | Norm./Act. |
|---|---|---|---|
| Input | 16x16x3 | | |
| Conv 3x3 | 16x16x32 | 1 | ReLU |
| Maxpool 2x2 | 8x8x32 | 2 | |
| Conv 3x3 | 8x8x64 | 1 | GN(4)/ReLU |
| Maxpool 2x2 | 4x4x64 | 2 | |
| Conv 3x3 | 4x4x128 | 1 | GN(8)/ReLU |
| Maxpool 2x2 | 2x2x128 | 2 | |
| Conv 3x3 | 2x2x256 | 1 | GN(16)/ReLU |
| Maxpool 2x2 | 1x1x256 | 2 | |

**GNN**

| Layer | Resolution | Norm./Act. |
|---|---|---|
| Input | $N$x256 | |
| Global Maxpool | 256 | |
| Linear (no bias) | 512 | ReLU |
| Linear (no bias) | 512 | ReLU |
| Linear | 1 | |

Table 3: The architecture of GNN and image patch encoder used in the experiments on Multi-MNIST. $N$ denotes the number of nodes in a graph. Note that we use an empty graph here.

| Layer | Resolution | Stride | Norm./Act. |
|---|---|---|---|
| Input | 54x54x3 | | |
| Conv 3x3 | 54x54x64 | 1 | BN/ReLU |
| Conv 3x3 | 18x18x64 | 3 | BN/ReLU |
| Conv 3x3 | 18x18x128 | 1 | BN/ReLU |
| Conv 3x3 | 6x6x128 | 3 | BN/ReLU |
| Conv 3x3 | 6x6x256 | 1 | BN/ReLU |
| Flatten | 9216 | | |
| Linear (no bias) | 512 | | ReLU |
| Linear (no bias) | 512 | | ReLU |
| Linear | 1 | | |

Table 4: The architecture of the plain CNN used in the experiments on Multi-MNIST.

| Layer | Resolution | Stride | Norm./Act. | | Layer | Resolution | Stride | Norm./Act. |
|---|---|---|---|---|---|---|---|---|
| **Key encoder** | | | | | **Value encoder** | | | |
| Input | 6x6x256 | | | | Input | 6x6x256 | | |
| Conv 1x1 | 6x6x256 | 1 | GN(4)/ReLU | | Conv 1x1 | 6x6x256 | 1 | GN(4)/ReLU |
| Conv 1x1 | 6x6x64 | 1 | | | Conv 1x1 | 6x6x256 | 1 | GN(4)/ReLU |
| **Query encoder** | | | | | **Post-attention Encoder** | | | |
| Input | 6x6x256 | | | | Input | 6x6x256 | | |
| Conv 1x1 | 6x6x256 | 1 | GN(4)/ReLU | | Conv 1x1 | 6x6x256 | 1 | GN(4)/ReLU |
| Conv 1x1 | 6x6x64 | 1 | | | Conv 1x1 | 6x6x256 | 1 | GN(4)/ReLU |

Table 5: The architecture of the relational module (4 heads) of relation netwrok used in the experiments on Multi-MNIST.

| Method | Train Acc | Test Acc |
|---|---|---|
| CNN(flatten) | 92.0(1.7) | 30.7(4.9) |
| CNN(max pooling) | 90.3(1.5) | 20.2(3.6) |
| CNN(sum pooling) | 91.0(0.9) | 5.7(3.4) |

(a) Multi-MNIST-CIFAR

| Method | Train Acc | Test Acc |
|---|---|---|
| CNN(flatten) | 90.5(2.9) | 12.0(2.1) |
| CNN(max pooling) | 83.1(0.7) | 9.74(0.4) |
| CNN(sum pooling) | 84.2(0.9) | 10.2(1.8) |

(b) Multi-MNIST-ImageNet

Table 6: Quantitative results of different design choices for the plain CNN on Multi-MNIST. The average with the standard deviation (in the parentheses) over 5 trials is reported.

# 4 FallingDigit

## 4.1 Environment Details

The foreground (digit) images are randomly selected from the MNIST dataset. For each digit, we only use one fixed image instance. The background images are either black or random selected from a subset in CIFAR-10 [10] dataset, and this subset contains 100 random selected images. All the foreground and background images are shared across the training and test environments.

## 4.2 Demonstration Acquisition

We train a CNN-based DQN to acquire the teacher policy, which is used to interact with the FallingDigit environment with three target digits to collect the demonstration dataset. During the interaction, we use the greedy policy derived from the $Q$ function, i.e., $\pi(s) = \arg\max_a Q(s, a)$. The demonstration dataset includes 60,000 images and each image is labeled with $Q(s, a)$ for all actions calculated by the teacher policy.

## 4.3 Self-supervised Object Detector

The self-supervised object detector is trained on the collected demonstration dataset. For the glimpse of each object, we apply the STN to crop a patch from the image according to the bounding box and resize it to 16x16. Table 7 shows the architecture of the object detector used in the experiments. For FallingDigit-Black, the background encoder and decoder are removed. Table 8 shows the hyperparameters of the object detector.

**Foreground Image Encoder**

| Layer | Resolution | Stride | Norm./Act. |
|---|---|---|---|
| Input | 128x128x3 | | |
| Conv 3x3 | 128x128x16 | 1 | BN/ReLU |
| Maxpool | 64x64x16 | 2 | |
| Conv 3x3 | 64x64x32 | 1 | BN/ReLU |
| Maxpool | 32x32x32 | 2 | |
| Conv 3x3 | 32x32x64 | 1 | BN/ReLU |
| Maxpool | 16x16x64 | 2 | |
| Conv 3x3 | 16x16x128 | 1 | BN/ReLU |
| Conv 1x1 | 16x16x128 | 1 | BN/ReLU |
| Conv 1x1 | 16x16x128 | 1 | BN/ReLU |
| | 16x16x1 (object presence $z^{pres}$) | 1 | Sigmoid |
| Conv 1x1 | 16x16x4 (bounding box mean $z^{where}$) | 1 | |
| | 16x16x4 (bounding box stdev $z^{where}$) | 1 | Softplus |

**Glimpse Encoder**

| Layer | Resolution | Stride | Norm./Act. |
|---|---|---|---|
| Input | 16x16x3 | | |
| Conv 1x1 | 16x16x16 | 1 | ReLU |
| Maxpool | 8x8x32 | 2 | |
| Conv 1x1 | 8x8x32 | 1 | ReLU |
| Maxpool | 4x4x32 | 2 | |
| Conv 1x1 | 4x4x64 | 1 | ReLU |
| Maxpool | 2x2x64 | 2 | |
| Conv 1x1 | 2x2x128 | 1 | ReLU |
| Maxpool | 1x1x128 | 2 | |
| Linear | 50 (mean $z^{what}$) | | |
| | 50 (stdev $z^{what}$) | | Softplus |

**Glimpse Decoder**

| Layer | Resolution | Stride | Norm./Act. |
|---|---|---|---|
| Input | 1x1x50 | | |
| Deconv 2x2 | 2x2x128 | 2 | ReLU |
| Conv 1x1 | 2x2x64 | 1 | ReLU |
| Deconv 2x2 | 4x4x64 | 2 | ReLU |
| Conv 1x1 | 4x4x32 | 1 | ReLU |
| Deconv 2x2 | 8x8x32 | 2 | ReLU |
| Conv 1x1 | 8x8x16 | 1 | ReLU |
| Upsample 2x2 | 16x16x16 | 2 | |
| Conv 1x1 | 16x16x4 | 1 | |

**Background Image Encoder**

| Layer | Resolution | Stride | Norm./Act. |
|---|---|---|---|
| Input | 128x128x3 | | |
| Conv 3x3 | 128x128x32 | 1 | BN/ReLU |
| Maxpool 2x2 | 64x64x32 | 2 | |
| Conv 3x3 | 64x64x32 | 1 | BN/ReLU |
| Maxpool 2x2 | 32x32x32 | 2 | |
| Conv 3x3 | 32x32x32 | 1 | BN/ReLU |
| Maxpool 2x2 | 16x16x32 | 2 | |
| Conv 3x3 | 16x16x32 | 1 | BN/ReLU |
| Maxpool 2x2 | 8x8x32 | 2 | |

**Background Image Decoder**

| Layer | Resolution | Stride | Norm./Act. |
|---|---|---|---|
| Input | 8x8x32 | | |
| Deconv 2x2 | 16x16x32 | 2 | BN/ReLU |
| Conv 3x3 | 16x16x32 | 1 | BN/ReLU |
| Deconv 2x2 | 32x32x32 | 2 | BN/ReLU |
| Conv 3x3 | 32x32x32 | 1 | BN/ReLU |
| Deconv 2x2 | 64x64x32 | 2 | BN/ReLU |
| Conv 3x3 | 64x64x32 | 1 | BN/ReLU |
| Upsample 2x2 | 128x128x32 | 2 | |
| Conv 3x3 | 128x128x32 | 1 | BN/ReLU |
| Conv 1x1 | 128x128x3 | 1 | |

Table 7: The architecture of the self-supervised object detector for all the experiments on FallingDigit.

| Name | Value | Schedule |
|---|---|---|
| max iteration | 100K | |
| optimizer | Adam | |
| batch size | 8 | |
| learning rate | 1e-3 | |
| gradient clip | 1.0 | |
| $z_{pres}$ prior | $0.1 \rightarrow 0.005$ | $0 \rightarrow 50K$ |
| $z_{pres}$ temperature | $2.5 \rightarrow 0.5$ | $0 \rightarrow 50K$ |
| $z_{where}$ prior mean | 0 | |
| $z_{where}$ prior stdev | 0.2 | |
| $z_{what}$ prior mean | 0 | |
| $z_{what}$ prior stdev | 1.0 | |
| $z_{what}$ dimension | 50 | |
| $z_{depth}$ prior mean | 0 | |
| $z_{depth}$ prior stdev | 1.0 | |
| $z_{depth}$ scale | 10.0 | |
| fg recon prior stdev | 0.15 | |
| bg recon prior stdev | 0.1 (Black) / 0.15 (CIFAR) | |

Table 8: The hyperparameters of the self-supervised object detector for all the experiments on FallingDigit. Note that the background module is disabled for FallingDigit-Black.

## 4.4 Policy GNN

**Network Architectures**   In the experiments on FallingDigit, the policy GNN is implemented as EdgeConv [20] in PyTorch Geometric [6]. The input graph is a complete graph, i.e., the edge set is $\{(i,j)|i,j \in \{1..n\}\}$ including self-loops, where $i, j$ are node indices. Each node corresponds to a detected object and the node feature includes an embedded image feature $x_{img}$ and the bounding box of the object $x_{box}$. To get $x_{img}$, we crop an image patch from the original image according to the bounding box, and resize it to 16x16, and then encode it by an image patch encoder. We concatenate $x_{img}$ and $x_{box}$ to get the node features and pass them into the GNN. The edge feature is the concatenation of the feature of the sender node, and the difference between the features of the sender and receiver nodes. Table 9 shows the architecture of GNN and image patch encoder used in FallingDigit experiments.

**Hyperparameters in Refactorization**   When training the GNN, the batch size is 64. The initial learning rate is 0.001, and is divided by 2 every 100K gradient updates. The network is trained with the Adam optimizer for 200K gradient updates.

**Image Patch Encoder**

| Layer | Resolution | Stride | Norm./Act. |
|---|---|---|---|
| Input | 16x16x3 | | |
| Conv 3x3 | 16x16x16 | 1 | ReLU |
| Maxpool 2x2 | 8x8x16 | 2 | |
| Conv 3x3 | 8x8x32 | 1 | ReLU |
| Maxpool 2x2 | 4x4x32 | 2 | |
| Conv 3x3 | 4x4x64 | 1 | GN(4)/ReLU |
| Maxpool 2x2 | 2x2x64 | 2 | |
| Conv 3x3 | 2x2x128 | 1 | GN(8)/ReLU |
| Maxpool 2x2 | 1x1x128 | 2 | |

**GNN**

| Layer | Resolution | Norm./Act. |
|---|---|---|
| Input | $N$x(128+4) | |
| Message Passing | $E$x(128+4)x2 | |
| Linear | $E$x128 | GN(8)/ReLU |
| Linear | $E$x128 | GN(8)/ReLU |
| Max Aggregation | $N$x128 | |
| Linear | $N$x128 | GN(8)/ReLU |
| Linear | $N$x128 | GN(8)/ReLU |
| Global Maxpool | 128 | |
| Linear | 128 | ReLU |
| Linear | 128 | ReLU |
| Linear | 3 | |

Table 9: The architecture of GNN and image patch encoder used in FallingDigit. $N$ denotes the number of nodes in a graph, and $E$ denotes the number of edges in a graph. We use complete graph here.

## 4.5 Baselines

**Network Architectures**   The architecture of plain CNN is illustrated in Table 10. For the Relation Net [21], we follow most of the design choices described in the original paper. We add a relational module at the 8x8 feature map. The relation module is the same as that used in Multi-MNIST, except that the resolution is 8x8.

| Layer | Resolution | Stride | Norm./Act. |
|---|---|---|---|
| Input | 128x128x3 | | |
| Conv 3x3 | 128x128x16 | 1 | ReLU |
| Maxpool | 64x64x16 | 2 | |
| Conv 3x3 | 64x64x16 | 1 | ReLU |
| Maxpool | 32x32x16 | 2 | |
| Conv 3x3 | 32x32x32 | 1 | ReLU |
| Maxpool | 16x16x32 | 2 | |
| Conv 3x3 | 16x16x64 | 1 | ReLU |
| Maxpool | 8x8x64 | 2 | |
| Conv 3x3 | 8x8x128 | 1 | ReLU |
| Maxpool | 4x4x128 | 2 | |
| Conv 1x1 | 4x4x128 | 1 | ReLU |
| Global Maxpool | 128 | | |
| Linear | 256 | | ReLU |
| Linear | 3 | | ReLU |

Table 10: The architecture of plain CNN used in FallingDigit.

**Hyperparameters in Tranining**   We use DQN [15] to train all the baselines. The related hyperparameters are listed in the Table 11.

| Name | Value | Schedule |
|---|---|---|
| max iteration | 10M | |
| optimizer | Adam | |
| learning rate | 1e-4 | |
| gradient clip | 10.0 | |
| $\epsilon$-greedy | $1.0 \rightarrow 0.1$ | $0 \rightarrow 300K$ |
| image normalizer | divide by 255 | |
| stacked frames | 1 | |
| target net update frequency | 500 steps | |
| replay buffer size | 100K | |
| discount factor | 0.99 | |
| training frequency | 4 steps | |
| batch size | 32 | |
| double Q | Yes | |

Table 11: The hyperparameters for training DQN on FallingDigit.

## 4.6 Evaluation Method

We train our GNN-based policy and all the baselines on the environment with three target digits and test them on the environments with more target digits. When evaluating all the policies, we take the best action suggested by the policy, i.e., $\pi_{eval}(s) = \arg\max_a Q(s, a)$. Since the environments are stochastic (the positions of all digits are random), we evaluate every policy on every environment for 1000 episodes and calculate the mean episode reward.

# 5 BigFish

## 5.1 Environment Details

Different from the original BigFish game, we modify the source code to add new background images to create test environments. To be more specific, we use the "Background-1.png", "Background-2.png", "Background-3.png" and "Background-4.png" from the "space-backgounds" directory in the ProcGen [3] source code. These background images are shown in Fig 1.

Figure 1: Background images used in BigFish test environments.

## 5.2 Demonstration Acquisition

To make our RL agent perform well in the test environment, it is important to train it on a diverse set of states which can cover most of states that might be encountered in the environments. In the BigFish environment, we found that if we simply use the best action suggested by the teacher policy, the demonstration dataset would only cover the states along the optimal trajectories, which are only a small portion of all feasible states. For example, in the optimal trajectories, the player will never get too close to the fishes that are bigger than the player itself. But in the testing environment, this may happen and our agent needs to know how to react in these states.

Therefore, we introduce $\epsilon$-greedy exploration [15] to increase the diversity of states. Since our teacher policy is trained by PPO [19], the output of the teacher policy is a categorical distribution over 15 discrete actions. When we apply $\epsilon$-greedy strategy, the agent will take a random action with probability $\epsilon$, otherwise, the agent will take a action suggested by the teacher policy (sampled from the categorical distribution). We collect demonstration from every level in the training set (level 0 to 200), and for each level we run the teacher policy multiple times.

We combine the datasets collected with different values of $\epsilon$ and the details are listed in Table 12. Since we run the experiments several times and collect several demonstration datasets, the size of the datasets vary. But we report a rough number of the sizes. The size of our combined demonstration dataset is around 800K.

| $\epsilon$ | # of trials in each level | dataset size |
|---|---|---|
| 0.5 | 5 | $\sim 350k$ |
| 0.3 | 3 | $\sim 300k$ |
| 0 | 1 | $\sim 150k$ |

Table 12: Composition of the demonstration dataset for BigFish

## 5.3 Self-supervised Object Detector

The self-supervised object detector is trained on a subset of the collected demonstration dataset, which consists of 60,000 images. The setting is similar to that of Multi-MNIST. For the glimpse of each object, we apply the STN to crop a patch from the image according to the bounding box and resize it to 8x8. Table 13 shows the architecture of the object detector used in the experiments. Table 14 shows the hyperparameters of the object detector.

**Foreground Image Encoder**

| Layer | Resolution | Stride | Norm./Act. |
|---|---|---|---|
| Input | 64x64x3 | | |
| Conv 3x3 | 64x64x64 | 1 | BN/ReLU |
| Maxpool 2x2 | 32x32x64 | 2 | |
| Conv 3x3 | 32x32x128 | 1 | BN/ReLU |
| Maxpool 2x2 | 16x16x128 | 2 | |
| Conv 3x3 | 16x16x256 | 1 | BN/ReLU |
| Maxpool 2x2 | 8x8x256 | 2 | |
| Conv 1x1 | 8x8x256 | 1 | BN/ReLU |
| Conv 1x1 | 8x8x256 | 1 | BN/ReLU |
| | 8x8x1 (object presence $z^{pres}$) | 1 | Sigmoid |
| Conv 1x1 | 8x8x4 (bounding box mean $z^{where}$) | 1 | |
| | 8x8x4 (bounding box stdev $z^{where}$) | 1 | Softplus |

**Glimpse Encoder**

| Layer | Resolution | Norm./Act. |
|---|---|---|
| Input | 8x8x3 | |
| Flatten | 192 | |
| Linear | 256 | GN(16)/ReLU |
| Linear | 256 | GN(16)/ReLU |
| Linear | 32 | |
| | 32 | Softplus |

**Glimpse Decoder**

| Layer | Resolution | Norm./Act. |
|---|---|---|
| Input | 32 | |
| Linear | 256 | GN(16)/ReLU |
| Linear | 256 | GN(16)/ReLU |
| Linear | 256 | Sigmoid |
| Reshape | 8x8x4 | |

**Background Image Encoder**

| Layer | Resolution | Stride | Norm./Act. |
|---|---|---|---|
| Input | 64x64x3 | | |
| Conv 3x3 | 64x64x64 | 1 | BN/ReLU |
| Maxpool 2x2 | 32x32x64 | 2 | |
| Conv 3x3 | 32x32x128 | 1 | BN/ReLU |
| Maxpool 2x2 | 16x16x128 | 2 | |
| Conv 3x3 | 16x16x256 | 1 | BN/ReLU |
| Maxpool 2x2 | 8x8x256 | 2 | |
| Conv 1x1 | 8x8x256 | 1 | BN/ReLU |

**Background Image Decoder**

| Layer | Resolution | Norm./Act. |
|---|---|---|
| Input | 256 | |
| Linear | 256 | BN/ReLU |
| Linear | 256 | BN/ReLU |
| Linear | 12288 | Sigmoid |
| Reshape | 64x64x3 | |

Table 13: The architecture of the self-supervised object detector for all the experiments on BigFish.

| Name | Value | Schedule |
|---|---|---|
| max iteration | 100K | |
| optimizer | Adam | |
| batch size | 32 | |
| learning rate | 1e-3 | |
| gradient clip | 1.0 | |
| $z_{pres}$ prior | $0.15 \rightarrow 0.05$ | $10K \rightarrow 50K$ |
| $z_{pres}$ temperature | $2.5 \rightarrow 0.5$ | $10K \rightarrow 50K$ |
| $z_{where}$ prior mean | 0 | |
| $z_{where}$ prior stdev | 0.3 | |
| $z_{what}$ prior mean | 0 | |
| $z_{what}$ prior stdev | 1.0 | |
| $z_{what}$ dimension | 32 | |
| $z_{depth}$ prior mean | 0 | |
| $z_{depth}$ prior stdev | 1.0 | |
| $z_{depth}$ scale | 10.0 | |
| fg recon prior stdev | 0.15 | |
| bg recon prior stdev | 0.15 | |

Table 14: The hyperparameters of the self-supervised object detector for all the experiments on BigFish.

## 5.4 Data Augmentation

We find that data augmentation is helpful to the generalization of object-centric GNN policy in the BigFish environment. Specifically, the detection threshold in SPACE is usually set 0.1 in this paper. But here we randomly select 30% of the object proposals (around 18 proposals) the has lower confidence score than the detection threshold and add them to the detection results. This data augmentation trick makes our object-centric GNN policy more robust to the false positive detections in the test environments. And this kind of data augmentation is not feasible for the CNN and Relation Net baselines.

## 5.5 Policy GNN

**Network Architectures**   In the BigFish experiment, the policy GNN is implemented as Edge-Conv [20] in PyTorch Geometric [6]. The input graph is a complete graph, i.e., the edge set is $\{(i,j)|i,j \in \{1..n\}, i \neq j\}$, where $i, j$ are node indices. Each node corresponds to a detected object and the node feature includes an embedded image feature $x_{img}$ and the position of the object $x_{pos}$. To get $x_{img}$, we crop a 12x12 image patch from the original image according to the position of the object and encode it by an image patch encoder. And the $x_{pos}$ is a 2-dim vector which represents the coordinates of the center of the object bounding box. We concatenate $x_{img}$ and $x_{pos}$ to get the node features and pass them into the GNN. The edge feature is the concatenation of the feature of the sender node, and the difference between the features of the sender and receiver node. Table 15 shows the detailed architecture of GNN and image patch encoder used in BigFish experiments.

| Image Patch Encoder | | | | | GNN | | |
|---|---|---|---|---|---|---|---|
| Layer | Resolution | Stride | Act. | | Layer | Resolution | Act. |
| Input | 12x12x3 | | | | Input | $N$x(512+2) | |
| Conv 3x3 | 10x10x32 | 1 | ReLU | | Message Passing | $E$x(514x2) | ReLU |
| Conv 3x3 | 8x8x64 | 1 | ReLU | | Linear | $E$x1024 | ReLU |
| Conv 8x1 | 1x8x64 | 1 | ReLU | | Linear | $E$x512 | |
| Flatten | 512 | | | | Max Aggregation | $N$x512 | |
| Linear | 512 | | ReLU | | Global Maxpool | 512 | |
| Linear | 512 | | | | Linear | 15 | |

Table 15: The architecture of GNN and image patch encoder used in BigFish. $N$ denotes the number of nodes in a graph, and $E$ denotes the number of edges in a graph. We use complete graph here.

**Hyperparameters in Refactorization**   When training the GNN, the batch size is 128. The initial learning rate is 8e-4, reduced to 8e-5 at the 560K-th gradient updates, and then reduced to 8e-6 at the 750K-th gradient updates. The network is trained with the Adam optimizer for 1150K gradient updates.

## 5.6 Baselines

**Network Architectures**   The CNN baseline is implemented according to the CNN architecture used in IMPALA [5], which is suggested by the ProcGen paper [3]. For the Relation Net [21] baseline, we use the same convolutional layers with IMPALA CNN, except we concatenate the spatial coordinates to the feature map as described in [21]. Then, we add a relational module after the final feature map by a residual connection. The architecture of the relation module is illustrated in Table 16. The output module of Relation Net is a flatten operator followed by a 2-layer MLP with hidden units of 256.

**Hyperparameters in Training**   We use PPO [19] to train all the baselines and use the same hyperparameters with the ProcGen paper [3], except that we use the easy mode of the game and train 200M frames.

## 5.7 Evaluation Method

We train our GNN-based policy and all the baselines on level 0-199 and test them on level 200- 399. When evaluating all the policies, we take the best action suggested by the policy, i.e., $\pi_{eval}(s) =$

| Layer | Resolution | Stride | Norm./Act. | | Layer | Resolution | Stride | Norm./Act. |
|---|---|---|---|---|---|---|---|---|
| **Key encoder** | | | | | **Value encoder** | | | |
| Input | 8x8x64 | | | | Input | 8x8x64 | | |
| Conv 1x1 | 8x8x64 | 1 | LN/ReLU | | Conv 1x1 | 8x8x64 | 1 | LN/ReLU |
| Conv 1x1 | 8x8x64 | 1 | | | Conv 1x1 | 8x8x64 | 1 | LN/ReLU |
| **Query encoder** | | | | | **Post-attention Encoder** | | | |
| Input | 8x8x64 | | | | Input | 8x8x64 | | |
| Conv 1x1 | 8x8x64 | 1 | LN/ReLU | | Conv 1x1 | 8x8x64 | 1 | LN/ReLU |
| Conv 1x1 | 8x8x64 | 1 | | | Conv 1x1 | 8x8x64 | 1 | LN/ReLU |

Table 16: The architecture of the relational module of Relation Net used in the experiments on BigFish. LN indicates Layer Normalization [1].

$\arg\max_a \pi(a|s)$ , instead of sampling from the categorical distribution. We evaluate every policy on every level from 200 to 399 once and calculate the mean episode reward. Since the environment is deterministic given the level index, and the polices are also deterministic by taking argmax, evaluating once is sufficient.

# 6 Pacman

## 6.1 Demonstration Acquisition

We train a CNN-based DQN to acquire the teacher policy, which is used to interact with the Pacman environment with two dots (food) to collect the demonstration dataset. During the interaction, we use the greedy policy derived from the $Q$ function, i.e., $\pi(s) = \arg\max_a Q(s, a)$. The demonstration dataset includes 60,000 images and each image is labeled with $Q(s, a)$ for all actions calculated by the teacher policy. According to our experiment results, this demonstration dataset is good enough for learning a GNN-based student policy which can generalize to the environments with more dots.

## 6.2 Self-supervised Object Detector

The self-supervised object detector is trained on the collected demonstration dataset. The setting is similar to that of Multi-MNIST. For the glimpse of each object, we apply the STN to crop a patch from the image according to the bounding box and resize it to 8x8. Table 17 shows the architecture of the object detector used in the experiments. Table 18 shows the hyperparameters of the object detector.

## 6.3 Policy GNN

**Network Architectures**   In the experiments on Pacman, the policy GNN is implemented as Point-Conv [16] in Pytorch Geometric [6]. The input graph is a complete graph, i.e., the edge set is $\{(i, j)|i, j \in \{1..n\}\}$ including self-loops, where $i, j$ are node indices. Each node corresponds to a detected object and the node feature includes an embedded image feature $x_{img}$ and the bounding box of the object $x_{box}$. To get $x_{img}$, we crop an image patch from the original image according to the bounding box, and resize it to 8x8, and then encode it by an image patch encoder. We concatenate $x_{img}$ and $x_{box}$ to get the node features and pass them into the GNN. The edge feature is the concatenation of the features of the sender node, and the difference between the bounding box position and size of the sender and receiver node. Table 19 shows the architecture of GNN and image patch encoder used in Pacman experiments.

**Hyperparameters in Refactorization**   When training the GNN, the batch size is 64. The initial learning rate is 0.001, and is divided by 2 every 100K gradient updates. The network is trained with the Adam optimizer for 500K gradient updates.

**Foreground Image Encoder**

| Layer | Resolution | Stride | Norm./Act. |
|---|---|---|---|
| Input | 64x64x3 | | |
| Conv 3x3 | 64x64x32 | 1 | BN/ReLU |
| Conv 2x2 | 32x32x32 | 2 | BN/ReLU |
| Conv 3x3 | 32x32x64 | 1 | BN/ReLU |
| Conv 2x2 | 16x16x128 | 2 | BN/ReLU |
| Conv 1x1 | 16x16x128 | 1 | BN/ReLU |
| Conv 1x1 | 16x16x128 | 1 | BN/ReLU |
| | 16x16x1 (object presence $z^{pres}$) | 1 | Sigmoid |
| Conv 1x1 | 16x16x4 (bounding box mean $z^{where}$) | 1 | |
| | 16x16x4 (bounding box stdev $z^{where}$) | 1 | Softplus |

**Glimpse Encoder**

| Layer | Resolution | Stride | Norm./Act. |
|---|---|---|---|
| Input | 8x8x3 | | |
| Conv 1x1 | 8x8x32 | 1 | GN(4)/ReLU |
| Maxpool | 4x4x32 | 2 | |
| Conv 1x1 | 4x4x64 | 1 | GN(4)/ReLU |
| Maxpool | 2x2x64 | 2 | |
| Conv 1x1 | 2x2x128 | 1 | GN(8)/ReLU |
| Maxpool | 1x1x128 | 2 | |
| Linear | 32 | | |
| | 32 | | Softplus |

**Glimpse Decoder**

| Layer | Resolution | Stride | Norm./Act. |
|---|---|---|---|
| Input | 1x1x32 | | |
| Deconv 2x2 | 2x2x128 | 2 | GN(8)/ReLU |
| Conv 1x1 | 2x2x64 | 1 | GN(4)/ReLU |
| Deconv 2x2 | 4x4x64 | 2 | GN(4)/ReLU |
| Conv 1x1 | 4x4x32 | 1 | GN(4)/ReLU |
| Deconv 2x2 | 8x8x32 | 2 | GN(4)/ReLU |
| Conv 1x1 | 8x8x16 | 1 | GN(4)/ReLU |
| Conv 1x1 | 8x8x4 | 1 | |

**Background Image Encoder**

| Layer | Resolution | Stride | Norm./Act. |
|---|---|---|---|
| Input | 64x64x3 | | |
| Conv 3x3 | 64x64x32 | 1 | BN/ReLU |
| Maxpool 2x2 | 32x32x32 | 2 | |
| Conv 3x3 | 32x32x32 | 1 | BN/ReLU |
| Maxpool 2x2 | 16x16x32 | 2 | |
| Conv 3x3 | 16x16x32 | 1 | BN/ReLU |
| Maxpool 2x2 | 8x8x32 | 2 | |
| Conv 3x3 | 8x8x32 | 1 | BN/ReLU |
| Maxpool 2x2 | 4x4x32 | 2 | |

**Background Image Decoder**

| Layer | Resolution | Stride | Norm./Act. |
|---|---|---|---|
| Input | 4x4x32 | | |
| Deconv 2x2 | 8x8x32 | 2 | BN/ReLU |
| Conv 1x1 | 8x8x32 | 1 | BN/ReLU |
| Deconv 2x2 | 16x16x32 | 2 | BN/ReLU |
| Conv 1x1 | 16x16x32 | 1 | BN/ReLU |
| Deconv 2x2 | 32x32x32 | 2 | BN/ReLU |
| Conv 1x1 | 32x32x32 | 1 | BN/ReLU |
| Deconv 2x2 | 64x64x32 | 2 | BN/ReLU |
| Conv 1x1 | 64x64x32 | 1 | BN/ReLU |
| Conv 1x1 | 64x64x3 | 1 | |

Table 17: The architecture of the self-supervised object detector for all the experiments on Pacman.

**Image Patch Encoder**

| Layer | Resolution | Stride | Norm./Act. |
|---|---|---|---|
| Input | 8x8x3 | | |
| Conv 3x3 | 8x8x32 | 1 | ReLU |
| Maxpool 2x2 | 4x4x32 | 2 | |
| Conv 3x3 | 4x4x64 | 1 | GN(4)/ReLU |
| Maxpool 2x2 | 2x2x64 | 2 | |
| Conv 3x3 | 2x2x128 | 1 | GN(8)/ReLU |
| Maxpool 2x2 | 1x1x128 | 2 | |

**GNN**

| Layer | Resolution | Norm./Act. |
|---|---|---|
| Input | $N$x(128+4) | |
| Message Passing | $E$x(128+4) | |
| Linear | $E$x128 | GN(8)/ReLU |
| Linear | $E$x128 | GN(8)/ReLU |
| Linear | $E$x4 | |
| Sum Aggregation | $N$x4 | |
| Global Maxpool | 4 | |

Table 19: The architecture of GNN and image patch encoder used in Pacman. $N$ denotes the number of nodes in a graph, and $E$ denotes the number of edges in a graph. We use complete graph here.

| Name | Value | Schedule |
|---|---|---|
| max iteration | 100K | |
| optimizer | Adam | |
| batch size | 8 | |
| learning rate | 1e-3 | |
| gradient clip | 1.0 | |
| $z_{pres}$ prior | $0.1 \rightarrow 0.005$ | $0 \rightarrow 50K$ |
| $z_{pres}$ temperature | $2.5 \rightarrow 0.5$ | $0 \rightarrow 50K$ |
| $z_{where}$ prior mean | 0 | |
| $z_{where}$ prior stdev | 0.2 | |
| $z_{what}$ prior mean | 0 | |
| $z_{what}$ prior stdev | 1.0 | |
| $z_{what}$ dimension | 32 | |
| $z_{depth}$ prior mean | 0 | |
| $z_{depth}$ prior stdev | 1.0 | |
| $z_{depth}$ scale | 10.0 | |
| fg recon prior stdev | 0.15 | |
| bg recon prior stdev | 0.15 | |

Table 18: The hyperparameters of the self-supervised object detector for all the experiments on Pacman.

## 6.4 Baselines

**Network Architectures**   The architecture of plain CNN is illustrated in Table 20. For the Relation Net [21], we follow most of the design choices described in the original paper. In our implementation, the input module of the Relation Net is the same as the convolutional layers used in the CNN baseline, except we concatenate the spatial coordinates to the feature map as described in [21]. Then we add a relational module after the final feature map by a residual connection. The architecture of the relation module is illustrated in Table 21. The output module of Relation Net is a feature-wise max pooling layer followed by a 2-layer MLP with hidden units of 256.

| Layer | Resolution | Stride | Norm./Act. |
|---|---|---|---|
| Input | 64x64x3 | | |
| Conv 3x3 | 64x64x16 | 1 | ReLU |
| Maxpool | 32x32x16 | 2 | |
| Conv 3x3 | 32x32x32 | 1 | ReLU |
| Maxpool | 16x16x32 | 2 | |
| Conv 3x3 | 16x16x64 | 1 | ReLU |
| Maxpool | 8x8x64 | 2 | |
| Conv 3x3 | 8x8x128 | 1 | ReLU |
| Maxpool | 4x4x128 | 2 | |
| Conv 1x1 | 4x4x128 | 1 | ReLU |
| Global Maxpool | 128 | | |
| Linear | 256 | | ReLU |
| Linear | 4 | | ReLU |

Table 20: The architecture of plain CNN used in Pacman.

**Hyperparameters in Tranining**   We use DQN [15] to train all the baselines. The related hyperparameters are listed in the Table 22.

## 6.5 Evaluation Method

We train our GNN-based policy and all the baselines on the environment with two dots and test them on the environments with more dots. When evaluating all the policies, we take the best action suggested by the policy, i.e., $\pi_{eval}(s) = \arg\max_a Q(s, a)$. Since the environments are stochastic (the positions of Pacman and dots are random), we evaluate every policy on every environment for 100 episodes and calculate the mean episode reward.

| Layer | Resolution | Stride | Norm./Act. |
|---|---|---|---|
| **Key encoder** | | | |
| Input | 7x7x64 | | |
| Conv 1x1 | 7x7x64 | 1 | LN/ReLU |
| Conv 1x1 | 7x7x64 | 1 | |
| **Query encoder** | | | |
| Input | 7x7x64 | | |
| Conv 1x1 | 7x7x64 | 1 | LN/ReLU |
| Conv 1x1 | 7x7x64 | 1 | |

| Layer | Resolution | Stride | Norm./Act. |
|---|---|---|---|
| **Value encoder** | | | |
| Input | 7x7x64 | | |
| Conv 1x1 | 7x7x64 | 1 | LN/ReLU |
| Conv 1x1 | 7x7x64 | 1 | LN/ReLU |
| **Post-attention Encoder** | | | |
| Input | 7x7x64 | | |
| Conv 1x1 | 7x7x64 | 1 | LN/ReLU |
| Conv 1x1 | 7x7x64 | 1 | LN/ReLU |

Table 21: The architecture of the relational module of Relation Net used in the experiments on Pacman. LN indicates Layer Normalization [1].

| Name | Value | Schedule |
|---|---|---|
| max iteration | 10M | |
| optimizer | Adam | |
| learning rate | 1e-4 | |
| gradient clip | 10.0 | |
| $\epsilon$-greedy | $1.0 \rightarrow 0.1$ | $0 \rightarrow 1M$ |
| image normalizer | divide by 255 | |
| stacked frames | 1 | |
| target net update frequency | 500 steps | |
| replay buffer size | 300K | |
| discount factor | 0.99 | |
| training frequency | 4 steps | |
| batch size | 32 | |
| double Q | Yes | |

Table 22: The hyperparameters for training DQN on Pacman.

# 7 Robustness Analysis

In this section, we analyze the robustness of our two-stage refactorization framework, taking Pacman environment as an example. And we compare our two-stage refactorization framework with the end-to-end one-stage reinforcement learning method to show our framework is more robust to the low-quality detectors.

## 7.1 Robustness w.r.t low-recall detectors

First, we conduct experiments to test how our refactorized GNN policy performs with a low-recall detector. Since the recall/AP of our object detector on Pacman is quite high, we randomly removed some detected objects to simulate the behaviours of a low-recall detector.

The detected objects are randomly removed in the demonstration dataset **during training** but are not removed **during testing**. We experiment with three different ratios of removed objects: 10%, 50% and 90%. Surprisingly, it is observed that even 50% objects are removed, the policy GNN can also imitate a reasonable policy from the demonstration dataset, and still generalizes well to the environments with more dots. We argue that it results from both the nature of the game itself and the robustness of our framework. Fig 2 illustrates the quantitative results.

In contrast, if we train our policy GNN with reinforcement learning (DQN) with 50% objects missing, it cannot converges to a reasonable good solution.

Figure 2: Quantitative experiments of robustness test on Pacman. We randomly remove 10%, 50% or 90% detected objects during training and report the test performance in the environments with different number of dots.

## 7.2 Robustness w.r.t low-precision detectors

Second, we test whether our refactorized GNN policy is robust to a low-precision detector. Similar to Sec 7.1, we simulate the behaviours of a low-precision detector by modifying a good detector. Specifically, we randomly select 25 object proposals with confidence scores lower than the threshold, which means they are not real objects, and add them to the detection results.

With such a low-precision detector, our refactorized GNN policy can still generalize well the environments with 10 dots (gets 8.29, averaged by 8 runs). In contrast, if we train our policy GNN with reinforcement learning (DQN) with this low-recall detector, the resulting GNN policy cannot consistently generalize to the environments with 10 dots (gets 4.43, averaged by 9 runs).

## 7.3 Summary

Through the above presented experiments, we find that our refactorized GNN is pretty robust w.r.t the low-recall detectors and low-precision detectors in the Pacman environment. In contrast, training the same object-centric GNN using reinforcement learning with low-quality detectors may lead to optimization or generalization problems. This is one of the reasons that we choose to break the policy learning problem into two stages instead of relying on end-to-end RL.