[Reviews · NeurIPS 2020]

Review 1

Summary and Contributions: This work focuses on compositional generalizability through learning an object-centric graph of the dynamics. They propose a self-supervised approach to learn object detectors. However, they argue that an end-to-end approach is naive and prone to errors, and so introduce a two-stage framework that first learns a teacher policy, then trains an unsupervised object detector to output proposals, with GNN to imitate the teacher policy. The authors call this teacher-student approach a "refactorization," and argue that it demonstrates better compositional generalization and interpretability.

Strengths: The improvement in performance over the teacher policy in Pacman and BigFish are impressive. The use of self-supervised object detection methods that don't rely on dynamics or the MDP seems novel, and empirically appears to work well in comparison to end-to-end graph network methods ([36]). I appreciate the thoughtful analysis in the conclusion, and commend the authors for pointing out the limitations of the method.

Weaknesses: Only one environment from ProcGen and Atari are showcased. One of the benefits of these benchmarks is the ability to test the robustness of your method on many different but related tasks. It would significantly strengthen your paper to include additional environments, even if performance is poor, if the performance can be explained. There should be ablations on the refactorization process, how important is that compared to the object proposal method? It seems that the other baselines should use a similar distillation process to be fair?

Correctness: Yes, they seem correct.

Clarity: There are some grammatical mistakes and misspellings, and there are sections and subsections with no text but just jump directly into a new section. There should be introductory comments to roadmap each section. What is DP in table 1? update: I now understand they are data parameters, but this should be clarified in the text. In line 159, do you mean CIFAR-task, not CIFAR-Recon? I assume CIFAR-Recon and ImageNet-Recon are features learned with an auto encoder? This needs to be defined explicitly in the text. Why are CIFAR-Task and ImageNet-Task using two different ways of computing the features, if they're just two different datasets for the same task/method? Why are there background features in ImageNet but not in CIFAR? The legends are too small in Fig 1 and should be made larger to see the colors better.

Relation to Prior Work: A similar work that also learns a GN with self-supervised object detectors is [1]. Given the comparison to relation networks, going more in depth into the algorithmic differences to account for the performance differences would be helpful. Is the drop in performance from relation networks to your method because of poor choice in nodes because the end-to-end method is worse at finding objects? 1. Contrastive Learning of Structured World Models - T. Kipf, E. van der Pol, M. Welling, ICLR 2020.

Reproducibility: Yes

Additional Feedback: I don't understand L165: "Second, we investigate how data parameters help generalization." It is unclear how this paragraph shows generalization? Were data parameters used for BigFish? Some of my concerns were answered satisfactorily in the rebuttal -- mainly an ablation that shows how much improvement is gained from refactorization. I am improving my score to a 7.


Review 2

Summary and Contributions: This paper proposes a two-stage framework to refactorize an overfitting CNN-based policy trained from image input into: (1) object proposals generation, (2) a generalizable GNN-based policy which takes objects of the image as input. In the second stage, GNN-based policy is cloned from CNN-based policy. The experimental results yield policies of better generalizability in unseen environments.

Strengths: One contribution is to down-weight incomplete proposals in GNN. Also the authors leverage the advantages of GNN, such as generalizability and interpretability. All three experimental results show GNN-based two-stage framework has better generalizability, which significantly reducing the overfitted CNN-based policy. The interesting one is that GNN can get better results with sparse/lost information, compared to CNN policy.

Weaknesses: From the idea level, this paper uses the existed methods, so its novelty is incremental. Why do we propose a two-stage framework, not an unified end-to-end policy learning based GNN? One reason (I guess) is that if we propose an end-to-end GNN policy model, we need to generate objects online because environment (or image) will change (depends what action is taken). This will be time-consuming.

Correctness: Yes

Clarity: Yes

Relation to Prior Work: Yes

Reproducibility: Yes

Additional Feedback: The result is good, especially when GNN policy clones the CNN policy with different domain inputs. The two-stage framework has high dependency on GNN and object proposals, which limit its application.


Review 3

Summary and Contributions: The paper proposes to improve the generalization of model-free RL agent by re-factorizing the previously trained policy network into an object-centric one with simple behavioral cloning. More specifically, the authors use an existing method (called SPACE) for unsupervised object discovery algorithms to obtain a pretrained object detector and extract object proposals from the scene. With these object proposals as inputs, a new policy network is trained to minimize the L2 prediction discrepancy between it and the original policy network. Further, the loss is re-weighted with a simple method such that images with poor object detection results contribute less to the total loss. The authors verify the effectiveness of their proposed approach on some hand-crafted environments and show that the learned feature representations are more task-centric, compared to those from purely reconstruction-based methods. In addition, the authors test their method on rather simple and customized pacman environment and show that the re-factored policy network can generalize better than the original one.

Strengths: 1. The paper demonstrates that unsupervisedly discovered object-centric representations are beneficial to out-of-distribution generalization. And the proposed two-staged distillation-like algorithm yields better model with no significant cost. 2. A data-dependent loss function is introduced to alleviate the negative effects of image samples with poor object detections results. 3. The paper shows that task-relevant representations are more natural and interpretable than reconstruction-based representations.

Weaknesses: 1. Some existing papers (e.g. [1]) have shown that object-centric representation generalizes well on novel scenes in Atari games. Given the environments in this paper are relatively simple and SPACE works quite well in such envs, the novelty of this paper is somewhat undermined. [1] Davidson, G. and Lake, B. M. (2020). Investigating simple object representations in model-free deep reinforcement learning. In Proceedings of the 42nd Annual Conference of the Cognitive Science Society.

Correctness: The claims and methods are correct.

Clarity: The rough idea of this paper is easy to grasp, though some parts are confusing. For instance, the abbreviation "DP" in Table 1 is unclear, if not completely unexplained.

Relation to Prior Work: Yes.

Reproducibility: Yes

Additional Feedback: 1. As mentioned before, what is "DP" in Table 1? The re-weighted loss? If it is, the authors need to explain it clearly in the results section. 2. How much does the reweighting of individual losses improve? Besides, a visualization of the most and least weighted data samples will be good.


Review 4

Summary and Contributions: The main focus of the paper is better generalization of RL policies for tasks which require graph reasoning. Their proposal is to first train a CNN based policy on the task, then to "refactor" that policy by using it as a teacher to train a GNN, in conjunction with a pre-trained object detector which feeds the GNN a graph. Edges are computed in a task-specific way. The idea is that for graph tasks the GNN has better generalization properties than the original (teacher) CNN but requires the structured input provided by the object detector in order to work well. The idea is explored on several visual tasks (e.g. summing mnist digits with varied backgrounds).

Strengths: -the refactorization is an interesting idea and seems to help -the results are positive and the approach is practical -by representing the state as a graph it enables better interpretation of the object features

Weaknesses: -it seems somewhat unlikely that the baselines chosen could really do as well as this approach since they are not given access to the object proposals. I wonder if there isn't some more reasonable baseline that could make use of that information to compare to the GNN? Could the object proposals be supplied as a kernel to the CNN of the relation network? -it wasn't totally clear to me why you needed to use the teacher policy. Does it not converge if you train the GNN with the (pre-trained) object detector from scratch? -how well does this work if you just use AIR with no background image. Being able to handle complicated background is certainly a positive thing but it is hard to tease out the difficulty of handling that "noise" robustly from the difficulty of generalizing to more objects. In fact, looking at table 1, it seems that the difference between your approach and the baselines decreases with the (presumbably) easier CIFAR backgrounds. Similarly with the less noisy BigFish backgrounds. -There is another line of work that has similar aims and uses a task that is similar to pac man that is relevant (but not cited): https://arxiv.org/abs/1609.05518 typos and suggestions: -line 62: I would use a different symbol than $\pi_i$ for this since $\pi$ is typically used for policies and this represents the output not the policy itself -line 68: necessary to have compositional generaliziabilty. generalizaiability -> generalizability - line128: on which some MNIST digtis digtis -> digits

Correctness: The method seems correct. One minor point, for table 1a, your results are shown in bold but are statistically the same as the CNN+DP.

Clarity: The paper is reasonably well written and I understood it (I think) without too much trouble.

Relation to Prior Work: Prior work is ok (see suggestion above for Garnelo paper)

Reproducibility: Yes

Additional Feedback: I have read the rebuttal and it largely addressed my concerns.

[Author Response · NeurIPS 2020]

**Major concerns:**

**To R2&R4 on end-to-end GNN policy learning, and robustness of our two-stage framework w.r.t. detection quality:** "end-to-end" represents the method that trains a GNN policy by RL directly, without refactorization. Firstly, the end-to-end method is not as robust as our two-stage framework. According to recent experiments, we can still generalize to 10 dots Pacman using a low recall detector ($50\%$ objects are missing) or a low precision detector (25 false positives are presented), with $\pm 8\%$ performance fluctuation. However, the end-to-end method suffers from noisy detections. With a low recall detector, it cannot converge in the training environment. With a low precision detector, its performance drops by $34\%$ and $46\%$ on 10 dots Pacman-CIFAR and Pacman-ImageNet, respectively. On BigFish, the end-to-end method fails to converge to a reasonable good solution in 200M steps. Secondly, it is more time-consuming to train and RL end-to-end with online object proposal generation. It takes $\sim 4.6\times$ time each training step, if we train a GNN policy by RL with proposal generation, instead of training a plain teacher CNN policy by RL. The training cost of refactorization step is negligible in comparison with the teacher policy training time.

**To R1 on ablations of refactorization:** As suggested, we train the Relation Net on the same demonstration dataset from a CNN teacher. For Pacman-CIFAR and Pacman-ImageNet, the mean episode reward in the environment with 10 dots increases to 79.05 and 82.21 respectively, close to ours (80.67 and 82.67). However, for BigFish, the Relation Net with refactorization got 27.18 in training set and only 15.68 in test set, significantly worse than ours. All the results are averaged by three runs. They prove that 1) refactorization is also useful for Relation Net; and 2) the advantages of our method come from *both the refactorization process and the detector+GNN architecture design*.

The two paragraphs above show that our two-stage training scheme can improve both the detector+GNN based framework and the Relation Net framework, in comparison to end-to-end training. Particularly, for the former (ours), the two-stage method gives a policy that is more robust to detector quality than end-to-end training. In principle, this two-stage scheme separates the difficulties in policy search (by RL) and in achieving generalizability by finding proper representation and inference method.

**To R3 on novelty:** Thanks for providing related works. However, most previous works either rely on symbolic inputs or GT objects, or only experiments on visually easy environments (no or very limited variation in fg/bg appearance). And the paper mentioned by R3 only qualitatively showed the consistence of the value function between prediction and humans at several handcrafted unseen states, without executing the policy in a new task. Differently, our settings are much harder, including complex games from public benchmarks like BigFish, and the generalizability is validated by executing the policy. We have to address them by a novel two-stage framework with self-supervised object proposals.

**Other concerns:**

**To R1 on more environments:** Thanks for the suggestion. Note that Atari does not provide a direct way to evaluate compositional generalizability studied in this paper since there lacks the targeted compositional variation. We plan to *customize and rewrite* more Atari games, and include more Procgen games, like StarPilot in the revision.

**To R1 on feature visualization:** L159 is correctly described; however, we will explain the terms more clearly in the revision. *Recon* refers to features from reconstruction (SPACE) and *Task* refers to those from task (Policy GNN). Features are computed in the same way for CIFAR-Task and ImageNet-Task. Our improved SPACE has few false positives (AP@0.25=92.4) on CIFAR, and thus the background cluster is too small to observe in the figure.

**To R1 on analysis on Relation Net:** Following the suggestion, we visualize the attention map of the Relation Net. We observe that although objects are implicitly attended to, the confidence in the attention map vary dramatically when the number of objects changes. It is inevitable if objects are not explicitly recognized.

**To R1, R3 on analysis on data parameters:** From Fig 2, we can observe that the images with missing detections are downweighted for training by data parameters. It prevents the model to fit a noisy (or insufficient) data point, which will in general hurts generalizability. Different from Multi-MNIST, good policies on Pacman and BigFish are not sensitive to missing detections. Thus, data parameters do not make much difference on the two games.

**To R4 on other baselines:** Our method gets the object proposals without additional information compared with the baselines. We do not know any other CNN-based baselines which could use the object proposals.

**To R4 on visual difficulty and generalization:** To tease out the difficulty of object detection, we apply our framework on Pacman without backgrounds. As expected by R4, our method has similar generalization performance to Relation Net, and is better than CNN (in the 10 dots Pacman environment, our method gets 70.2, Relation Net gets 69.6, CNN gets 12.9, averaged by three different runs). The gap between our method and Relation Net is much larger in the environments with more visual difficulty (Fig 3 in the paper). The comparison shows the robustness of our approach to visual difficulty, which aligns with our goal of achieving *compositional generalizability in more realistic environments with noisy detections*. Note that AIR can not extrapolate the number of objects, which inherently fails to generalize to environments with more objects.

**To R4 on related works and typos:** Thanks for providing the paper. We will cite it and correct typos in the revision.

[Meta-Review · NeurIPS 2020]

The paper generated discussion among the reviewers. On the positive side, there are strong empirical results, and the ablation study included in the rebuttal allowed the reviewers to see the merits of the proposed approach on top of SPACE. On the other hand, the novelty of the approach was considered low by a couple of the reviewers. However, the use of the approach in MDPs was judged interesting by the other two reviewers. On the balance, in my opinion, the paper does a good job of proposing an extension to existing work that is sufficiently novel and works very well, and the experiments are well done and convincing, so I am recommending acceptance. Please take into account the reviewers' comments in revising the paper.